# Dysplastic Nevi and Superficial Borderline Atypical Melanocytic Lesions: Description of an Algorithmic Clinico-Pathological Classification

**DOI:** 10.3390/dermatopathology12010003

**Published:** 2025-01-21

**Authors:** Sébastien Menzinger, Rastine Merat, Gürkan Kaya

**Affiliations:** 1Department of Dermatology and Venereology, University Hospital of Geneva, Rue Gabrielle-Perret-Gentil 4, 1205 Genève, Switzerland; 2Department of Clinical Pathology, University Hospital of Geneva, Rue Michel Servet 1, 1205 Genève, Switzerland

**Keywords:** dysplastic nevus, superficial borderline atypical melanocytic lesion, melanoma, dermatoscopy, clinico-pathological correlation, classification

## Abstract

The diagnosis, interpretation, and classification of melanocytic tumors is a very complex topic in the pathology and dermatopathology field that lacks standardization and is still subject to discordance and debate. Here, we review the definitions of dysplastic nevus and superficial atypical melanocytic proliferations and provide an overview of some areas still subject to debate and some attempts of standardization. Furthermore, we describe an algorithmic classification, and provide some examples of clinico-pathological correlation. This step-by-step algorithm has an educational purpose and may automatize the work of dermatopathologists. We hope that through further molecular studies, this fine-grained scheme will prove to be related to the biological behavior of these atypical melanocytic lesions.

## 1. Introduction

The clinico-pathological concept of dysplastic nevus (DN) was first described by Clark et al. in 1978, initially under the term “B-K mole syndrome”, in families with multiple melanocytic lesions having distinctive clinical and histological features associated with an increased risk of developing melanoma [1]. Since then, many names have been given to this “syndrome” and to the associated melanocytic lesions described in these patients, now known as DN, or “nevus with architectural disorder and cytological atypia of melanocytes” [2]. DN was then also described in nonfamilial and nonsyndromic settings [3]. Its very concept, relevance, and definition have been subject to controversy ever since. DN is often recognized as an important risk factor for melanoma, as some studies have shown a higher risk for people with DN exhibiting higher grades of melanocytic atypia [4,5,6,7].

Despite its name, DN is not considered as a true dysplastic precursor or premalignant lesion, in comparison, for example, with actinic keratosis. Nevertheless, melanoma can arise on a DN, but the individual/global risk seems to be very low [8,9,10,11].

In the fourth edition of the “WHO Classification of Skin Tumours” (2018), dysplastic nevi were defined as “a subset of melanocytic nevi that are clinically atypical and characterized histologically by architectural disorder and cytological atypia, always involving their junctional component. (…) Attribute of architectural disorder include the presence of junctional shoulders (lateral extension) adjacent to the dermal component (or the lesion may be entirely junctional), bridging of nests between adjacent elongated rete ridges, subtle suprabasal scatter of melanocytes confined to the lower epidermal levels (typically less pronounced than in melanoma), concentric and lamellar fibroplasia around elongated rete ridges, and a patchy lymphocytic infiltrate. Cytological atypia is characterized by enlargement of nuclei (with varying degrees of irregularity), chromatin clumping and hyperchromatism, and variably prominent nucleoli” [12].

The grading of the atypia in DN is also still a subject of debate. There is no established consensus, and the reliability is relatively poor among specialists [13]. Some classifications use only the cytonuclear atypia: nuclear size, variability in size and shape of nucleus, chromatin repartition, size of the nucleolus, and aspect of the cytoplasm [14]. Controversy exists regarding the use of architectural disorganization as a criterion for the grading. Nevertheless, some classifications also use architectural criteria to grade the atypia [4,15,16]; considering the following, for example: “lentiginous melanocytic hyperplasia (from discontinuous to confluent), upward melanocyte migration (little or no to fully pagetoid spread), nesting variation and bridging (few nests to confluent nests)” in addition to cytonuclear criteria [15]. Cytologic atypia is often graded as mild, moderate, or severe. The grading criteria published by the WHO in 2018 eliminated a “moderate” dysplasia category and simplified grading to low-grade dysplasia and high-grade dysplasia [12,17].

Another group of benign melanocytic proliferations is known to show a certain degree of cytological and architectural atypia, sometimes with features overlapping with dysplastic nevus, and is identified as “nevus of special site”. This nevus is not considered as a risk marker of melanoma, and is benign. It is important to recognize these nevi to avoid a misdiagnosis, and to prevent overtreatment of this kind of lesion. There are many sites on the body that are recognized as “special sites”, but the more frequently encountered are acral nevi, genital nevi, and flexural nevi [18].

Many terms exist for the categorization of melanocytic lesions with atypia (severe or not) that are ambiguous and defy proper categorization as clearly benign or clearly malignant, including MELTUMP (melanocytic tumors of uncertain malignant potential) and SAMPUS (superficial atypical melanocytic proliferations of uncertain significance). These denominations do not reflect distinct entities but have in common a degree of cytoarchitectural atypia exceeding what is acceptable for nevi, yet insufficient to justify a melanoma diagnosis [19]. Many entities could be classified in these two categories. Superficial borderline atypical melanocytic lesion can be classified as “SAMPUS”, as well as melanocytic acral nevus with intraepidermal ascent of cells (MANIAC), de novo intraepidermal epithelioid melanocytic dysplasia (DNIEMD), and eventually THIn Melanoma of Uncertain Metastatic Potential (THIMUMP) [20,21,22,23,24,25,26]. Such categorization raises questions on how to manage such lesions. Which excision margins are advisable [27]?

Because of the lack of standardization in the histologic diagnosis of melanocytic lesions, in 2014, Piepkorn et al [28]. developed the “Melanocytic Pathology Assessment Tool and Hierarchy for Diagnosis (MPATH-Dx)” schema. It is a diagnostic-treatment mapping tool that categorizes diverse nomenclature into a hierarchy of management interventions, with the aim of diminishing ambiguity in pathology reports and standardizing management options. They divided the reporting of melanocytic lesions into five categories that allow the classification of the lesions and the association of each category to a clinical outcome/attitude. DN with mild atypia were classified in class 1 (no apparent risk for continued local proliferation and adverse outcome), with moderate atypia in class 2 (low-level risk for local proliferation of remaining cells; probability of progressive disease unlikely; potentially adverse outcome in some cases), and with severe atypia in class 3 (higher likelihood of local tumor progression and greater need for intervention; no reliable data for actual risk), which was the same category as in situ melanoma. For class 3 lesions, for example, repeated excision was recommended with at least a 5 mm margin. Class 4 and 5 were for invasive melanomas (pT1a and T1b or more, respectively) [28]. We will further discuss below some studies that have evaluated this scheme and its reproducibility.

A “Revision of the Melanocytic Pathology Assessment Tool and Hierarchy for Diagnosis (MPATH-Dx V2) Classification Schema for Melanocytic Lesions” was published very recently. In this revision, the classification of the melanocytic lesions was proposed in four categories: I—low-grade, including DN with mild atypia; II—high-grade, including DN with severe atypia; III—melanoma pT1a; IV—melanoma ≥ pT1b. One of the reasons that led to this modification was the poor rates of the interobserver (as low as 25%) and intraobserver (as low as 35%) agreement of moderately atypical lesions in class II of the MPATH-Dx V1. The authors also decided to devise a new two-tiered classification schema (low-grade and high-grade) that replaces the old three-tiered system: class I, defined as mild atypia; class II, moderate atypia; and class III, severe atypia [29].

In the University Hospital of Geneva, Switzerland, we use a fine-grained method to classify dysplastic nevi and superficial borderline atypical melanocytic lesions. This classification is based on cytonuclear and architectural grounds and aims to allow the automatized categorization of lesions. Moreover, as illustrated below, we propose to use clinico-pathological correlation, particularly for ambiguous lesions. Our experience supports the hypothesis of an existing correlation between the observed clinical/dermatoscopic atypia and the reported degree of histological atypia. The algorithmic approach described in this article has an educational purpose for dermatopathologists and dermatologists using dermatoscopy in their routine practice.

## 2. Our Classification

Facing a superficial melanocytic proliferation, with only junctional, or with junctional and dermal component, we first determine whether the lesion shows atypia or not, in the junctional and dermal component. If junctional atypia is noted, we further determine if the lesion is “dysplastic” or not. Here, we use the same criteria as cited above (WHO): architectural disorder (shoulder sign in case of a compound lesion, bridging of nests between adjacent elongated rete ridges, disorganization of the proliferation), concentric and lamellar fibroplasia around elongated rete ridges, patchy lymphocytic infiltrate and presence of melanophages, and the presence of cytological atypia. We also use the same cytological, nuclear, and nucleolar criteria to define the grade of atypia, but the major difference is that we also use the architectural disorganization to define the grade of atypia, with more granularity. We classify atypia in five main categories: mild, moderate, distinct, marked, and severe. Sometimes a lesion can show foci with different patterns of architectural disorders and cytonuclear atypia, so ʺmild to moderateʺ, ʺmarked to severeʺ, etc., are also used as intermediate categories. To correctly apply this classification, particularly for lesions with a higher grade of atypia, an immunohistochemistry analysis with a melanocytic marker is also performed. We routinely use Melan-A/MART-1 staining in cases with a suspicion on the HE sections of marked to severe atypia, lentiginous proliferation, and pagetoid spread. Furthermore, if there are junctional and dermal atypia, we search for malignant melanoma criteria (cytological, architectural, immunohistochemical). It is noteworthy that melanocytic proliferations with mild to distinct junctional atypia never exhibit atypia in the dermal component. DN with mild to marked atypia can be classified as “low-grade” or class I in the MAPTH-Dx V2 classification schema. DN with severe atypia, even if these atypia are only focally observed, can be classified as “high-grade” or class II.

### 2.1. DN with Mild Atypia

DN with mild atypia is defined by mild cytological, nuclear, and nucleolar atypia, as in other classifications, and architecturally by a discrete junctional proliferation showing no nests or very small nests, irregularly distributed along the dermal–epidermal junction. These nevi frequently show lentigo-like epidermal hyperplasia and hyperpigmentation (Figure 1A,B).

### 2.2. DN with Moderate Atypia

DN with moderate atypia is also defined by mild cytological, nuclear, and nucleolar atypia, but architecturally by a more pronounced junctional proliferation showing well-developed nests, not completely filling the rete ridges (Figure 1C,D).

### 2.3. DN with Distinct Atypia

DN with distinct atypia is defined by more pronounced cytological, and nucleolar atypia, and architecturally by voluminous nests, or solitary units, completely filling the rete ridges. Both patterns can be frequently observed in the same lesion (solitary units filling the rete ridges and voluminous nests). There is no suprabasal melanocyte proliferation and no solitary units between the rete ridges (Figure 1E–H).

### 2.4. DN with Marked Atypia

DN with marked atypia is defined by marked cytological, nuclear, and nucleolar atypia, and architecturally by a continuous proliferation along the dermal–epidermal junction, or with solitary units between the rete ridges, eventually with the presence of few suprabasal melanocytes, but only focally (Figure 2A–C).

### 2.5. DN with Severe Atypia

DN with severe atypia is defined by marked (or severe) cytological, nuclear, and nucleolar atypia, a continuous proliferation along the dermal–epidermal junction, and the presence of more suprabasal melanocytes, generally restricted to the central part of the lesion. The lesion does not fulfill the criteria of a melanoma in situ. There is an overlap with DN with marked atypia, because the only difference is the presence of a more pronounced pagetoid spread. These lesions are categorized as DN with marked to severe atypia (Figure 2D–F).

### 2.6. Superficial Borderline Atypical Melanocytic Lesion

#### 2.6.1. With Junctional Atypia

These superficial borderline atypical melanocytic lesions are only junctional or compound lesions. In the case of a compound melanocytic lesion, the atypia are restricted to the junctional part. The lesion is usually characterized by severe cytonuclear and architectural atypia, but does not fulfill all the criteria of a melanoma in situ. However, the presence of these significant atypia does generally not allow pathologists to exclude a melanoma in situ, or the possibility of a melanoma in situ arising on a pre-existing nevus, usually dysplastic (Figure 2G–I). These criteria include the following:Cytoplasmic atypia: dirty cytoplasm, epithelioid morphology, vacuolated cytoplasm;Nuclear atypia: large nucleus, hyperchromasia;Nucleolar atypia: prominent nucleolus;Architectural atypia: disorganized/anarchic proliferation with irregular nests and numerous solitary units;Some pagetoid spread.

If all these criteria are observed throughout the lesion, the lesion fulfills the criteria of a melanoma in situ and should therefore not be considered in this category.

#### 2.6.2. With Junctional and Dermal Atypia

The same criteria apply to these lesions, i.e., the presence of marked to severe cytonuclear and architectural atypia, but without fulfillment for the criteria of a superficial spreading melanoma. In the dermal component, we may observe the presence of cells showing the same degree of atypia than in the junctional component, the absence of a clear maturation, and the presence of mitotic figures. These atypia may be present in part of or in all the cells of the dermal component (See Section 3.4).

It has been established that PRAME immunostaining has utility as an adjunct for evaluating challenging melanocytic proliferations [30]. We routinely use it for lesions showing marked atypia or more and recommend its use for all ambiguous lesions ranging from DN with marked atypia to atypical melanocytic lesions with junctional and dermal atypia. Its use is also recommended for lesions exhibiting different grades of atypia in different areas, and can lead to a diagnosis of melanoma arising on a pre-existing nevus. We believe that in the future, PRAME immunostaining should be integrated in the grading criteria of DN and superficial borderline atypical melanocytic lesions since diffuse and strong PRAME positivity is indicative of a higher grade of atypia.

Here, we emphasize the importance of incorporating architectural atypia in the grading classification. The architecture is a critical aspect in the definition of many melanocytic lesions, including melanomas. Why should it be different for defining the degree of atypia in DN? Moreover, similarly to cytological atypia, architectural atypia may reflect the degree of biological aberrations in the lesion.

## 3. Clinico-Pathological Correlation

We use systematic clinico-pathological correlation in our routine practice, as shown here in the provided iconography. In our experience, DN dermatoscopic features are often correlated to the degree of atypia found in the lesion. These dermatoscopic features are generally observed at least on part of the lesion.

### 3.1. DN with Mild Atypia

Histologically, DN with mild atypia frequently shows lentigo-like epidermal modifications. Clinically, these lesions usually exhibit a thin pigmented reticular network (Figure 3A–C).

### 3.2. DN with Moderate Atypia

These lesions show a slightly larger pigmented network and/or small globi, probably corresponding to the medium-sized nests observed histologically (Figure 3D–G).

### 3.3. DN with Distinct Atypia

These lesions usually show a larger and coarser pigmented network and/or larger globi, sometimes at the periphery of the lesion, probably representing the larger nests observed histologically (Figure 4).

### 3.4. DN with Marked and/or Severe Atypia, and Superficial Borderline Atypical Melanocytic Lesion

We usually do not observe major or specific differences in terms of clinical atypia between DN with marked and/or severe atypia and superficial borderline atypical melanocytic lesions. However, these lesions usually show at least one worrisome dermatoscopic feature such as an eccentric area without structure, radial striae, pseudopods, white lines more or less diffuse possibly generating an “inverted network”, abrupt interruption of the network, varying pattern, asymmetry, etc. (Figure 5, Figure 6, Figure 7 and Figure 8). Figure 9 and Figure 10 are a melanoma in situ and a microinvasive superficial spreading melanoma, shown for comparison.

The final assessment of a melanocytic lesion should remain integrative, and should include the clinical data, including the dermatoscopic picture, the evolution of the lesion, its topography, the age of the patient, etc., and the histopathological picture and the ancillary tests, including immunochemistry. Table 1 summarizes our classification and the clinical/dermatoscopic features usually observed.

## 4. Discussion

The purpose of this article was to share our grading classification of the atypia in DN and superficial borderline atypical melanocytic lesions with the medical and scientific community, and also to provide a non-exhaustive overview of the accepted classifications and areas still subject to debate. We are fully aware of the controversies surrounding this topic. Our objective was not to further complexify a debated classification, nor to demonstrate that our classification is more effective, but rather introduce it to the community of practitioners as a systematic classification tool. We think that adding clear categories and a step-by-step algorithm could “automatize” the work of dermatopathologists and generate reproducibility (Figure 11). Obviously, such reproducibility would need to be tested similarly to the way MPATH-Dx scheme version 1 was tested.

In 2016, the International Melanoma Pathology Study Group published an evaluation of the MPATH-Dx scheme. Sixteen dermatopathologists regarded as melanocytic lesion experts provided independent case-level diagnoses and treatment suggestions for 48 melanocytic lesions. Individual diagnoses were mapped to corresponding MPATH-Dx classes. Mean weighted kappa coefficients for diagnostic agreement following MPATH-Dx mapping (assuming least and most severe diagnoses, when necessary) were 0.70 and 0.72, respectively, while the correlation between MPATH-Dx categorization and treatment suggestions was 0.91 [31].

A 2020 survey study of 160 dermatopathologists found that nearly all participants (99%) had witnessed different terminologies used to describe the same melanocytic skin lesion, and viewed it as confusing to primary care physicians (98%) and frustrating to pathologists (83%). Most participants perceived that the uniform adoption of the MPATH-Dx would improve the communication with other pathologists and treating physicians (87%) [17,32].

A large study evaluating the accuracy and reproducibility in diagnosing melanocytic lesions using the MPATH-Dx schema was published in 2017. Reproducibility was assessed by intraobserver and interobserver concordance rates. The pathologists used diverse diagnostic terms to classify the melanocytic proliferations. The mean number of diagnostic terms applied for each case in phase 1 was 10. Concordance with the reference was higher in class I (92%) and V (72%), and lower in class II (25%), III (40%), and IV (43%). Intraobserver reproducibility was assessed for 118 pathologists based on two different phases of interpretation. Eight or more months separated the two phases, and the participating pathologists were blinded and not aware that these were the same cases. Cases interpreted in phase 1 as class I and class V were likely to receive a diagnosis in the same class when interpreted by the same pathologist in phase 2 (77% and 83%, respectively). Pathologists’ reproducibility was lower for cases initially interpreted as class II (35%), class III (60%), and class IV (63%) [33].

These results show that the extreme categories, i.e., class I, such as a common nevus, and class V (or class IV in the MPATH-Dx V2), such as a superficial spreading melanoma with a Breslow > 0.8 mm, are more easily diagnosed with a certain degree of reproducibility. The results also show that intermediate categories have a low inter- and intraobserver reproducibility, and that the use of broader categories does not facilitate an easier classification of these intermediate or ambiguous lesions.

One of the most important and determining factors is to define the follow-up and management of such ambiguous lesions, and to reach a consensus between experts. Table 2 summarizes the associated management for each of our categories with a comparison with MPATH-Dx V2. There are a few slight differences. It is usually recommended to perform a re-excision with 5–10 mm margin for ambiguous melanocytic tumors. However, we know that the utility of such re-excision has been questioned in various studies. A retrospective cohort study was conducted in the Netherlands in patients with a diagnosis of MELTUMP or SAMPUS, with a minimum follow-up period of 3 years (n = 2692). The results indicated that 1.3% (16/1248) of MELTUMP and 0.2% of SAMPUS demonstrated local recurrence, local progression to melanoma, or metastatic behavior [19]. These results clearly question the benefits of such re-excision procedures. Obviously, to clarify the management of these ambiguous lesions, further studies are needed. However, we believe that the management algorithm should ultimately be integrative and also based on the clinical suspicion level, which is best evaluated by dermatoscopic documentation. Here, we have illustrated this clinico-pathological correlation by providing the dermatoscopic images for each shown example of DN categories. We believe that since our classification takes into consideration the architecture of the proliferation and not only the cytonuclear atypia, a stronger correlation should emerge between the clinical/dermatoscopic atypia and the degree of histological atypia assessed by our classification scheme.

## 5. Conclusions and Future Directions

In this study, we describe an algorithmic classification of dysplastic nevi and superficial borderline atypical melanocytic proliferations, and provide examples of clinico-pathological correlation. Beyond its potential usefulness for routine practice in dermatopathology, we also emphasize that our classification scheme should prove useful in the discovery and evaluation of prognostic markers for melanoma, which are lacking in our arsenal at a time when the indications for adjuvant treatments are steadily expanding. Indeed, even though, as previously discussed, the risk of DN transformation into melanoma is extremely low, and these lesions cannot be considered pre-cancerous from an epidemiological standpoint, their variability in cytological, and especially architectural, phenotype, as captured by our classification scheme, necessarily reflects subtle biological differences. Our reasoning is as follows: if a marker can distinguish such subtle differences, it should, in principle, be able to distinguish malignant lesions whose phenotypic behavior is much more diverse in terms of metastatic potential. On the other hand, if a search for biological markers, as in the example of a study that sought to classify these lesions from a genetic standpoint [34], is applied to a simplified classification, the results will always be questionable due to the inability to establish a true “dose–response” effect for the investigated marker(s). Undoubtedly, we hope that the granularity of the classification proposed here will help to overcome these obstacles, and that through further molecular studies, this fine-grained scheme will prove to be related to the biological behavior of these atypical melanocytic lesions.

## Figures and Tables

**Figure 1 dermatopathology-12-00003-f001:**
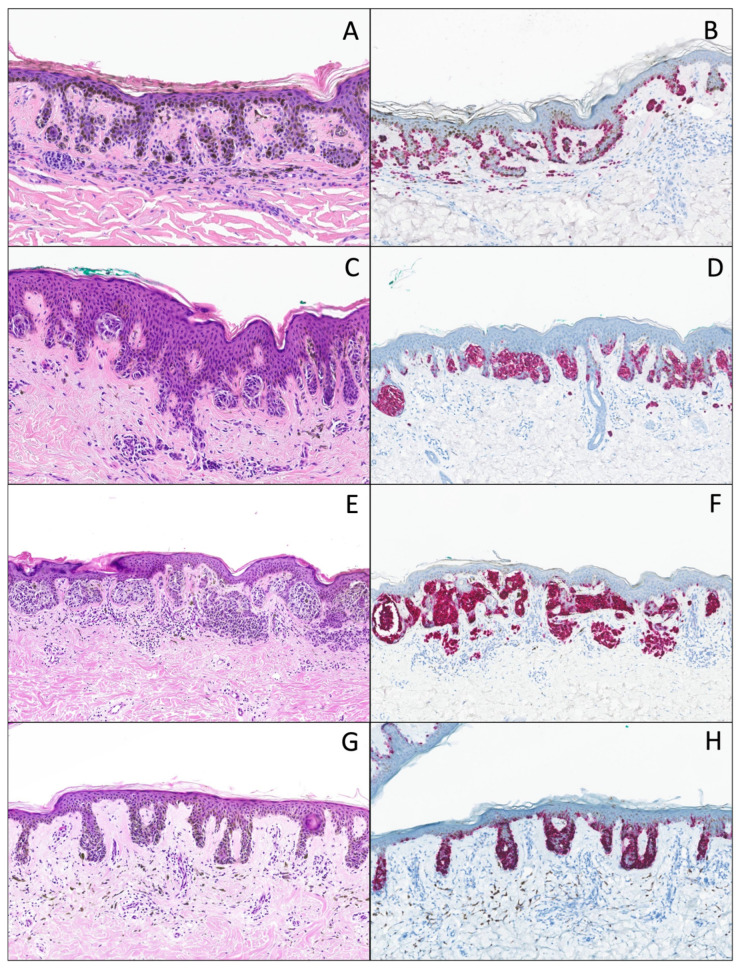
(**A**) Hematoxylin and eosin stain (original magnification 10x); DN with mild atypia. Pigmented parakeratosis overlying a hyperplastic epidermis with lentigo-like features, associated with a melanocytic proliferation with a discrete junctional proliferation showing very small nests, irregularly distributed along the dermal–epidermal junction. There is a slight “shoulder sign”, slight papillary fibroplasia, and a slight-to-moderate dermal lymphocytic infiltrate associated with melanophages. (**B**) Melan-A/MART-1 IHC (original magnification 10x) of the same lesion, showing the same features and the absence of continuous proliferation and pagetoid spread. (**C**) Hematoxylin and eosin stain (original magnification 10x); DN with moderate atypia. The junctional component shows irregularly distributed nests, but not completely filling the rete ridges, and mild cytonuclear atypia. There is a “shoulder sign”, papillary fibroplasia, and a slight dermal lymphocytic infiltrate associated with melanophages. (**D**) Melan-A/MART-1 IHC (original magnification 10x) of the same lesion, showing the same features and the absence of continuous proliferation and pagetoid spread. (**E**) Hematoxylin and eosin stain (original magnification 10x), DN with distinct atypia, “nested type”. The junctional component shows large irregularly distributed nests, completely filling the rete ridges, and moderate cytonuclear atypia. There is a “shoulder sign”, papillary fibroplasia, and a slight-to-moderate dermal lymphocytic infiltrate associated with melanophages. (**F**) Melan-A/MART-1 IHC (original magnification 10x) of the same lesion, showing the same features and the absence of continuous proliferation and pagetoid spread. (**G**) Hematoxylin and eosin stain (original magnification 10x); DN with distinct atypia; solitary unit type. Lentiginous hyperplasia and hyperpigmentation of the epidermis, and a junctional melanocytic proliferation with moderate cytonuclear atypia, filling the rete ridges is visible without signs of continuous proliferation or pagetoid spread. (**H**) Melan-A/MART-1 IHC (original magnification 10x) of the same lesion, showing the same features.

**Figure 2 dermatopathology-12-00003-f002:**
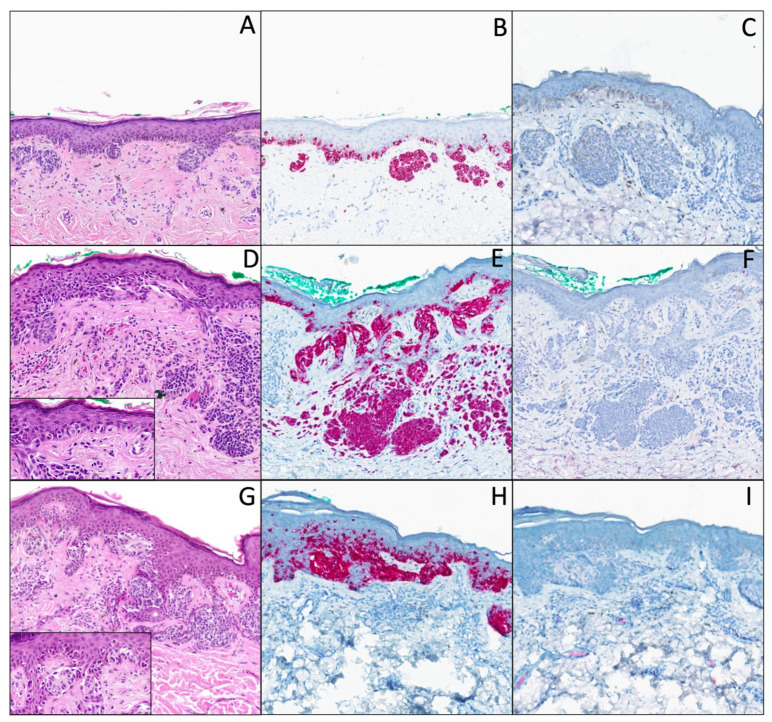
(**A**) Hematoxylin and eosin stain (original magnification 10x), DN with marked atypia. The junctional component shows marked cytonuclear atypia, in this case with marked nuclear hyperchromasia, and a continuous proliferation along the dermal–epidermal junction, with solitary units between the rete ridges. (**B**) Melan-A/MART-1 IHC (original magnification 10x), with very few suprabasal melanocytes. There is a “shoulder sign”, papillary fibroplasia, and a slight dermal lymphocytic infiltrate associated with melanophages. (**C**) PRAME IHC (original magnification 10x) is negative (note that this immunochemistry is not exactly performed on the same level as Melan-A). (**D**) Hematoxylin and eosin stain (original magnification 10x) DN with severe atypia. Junctional component shows a very disorganized proliferation with solitary units and irregular nests, with (inset) marked/severe cytonuclear atypia (abundant cytoplasm, and irregular nuclei with hyperchromasia; original magnification 40x). The dermal component shows maturation, without atypia, and with a superficial congenital architecture. The “shoulder sign” is not evident in these pictures, there is papillary fibroplasia, and a slight dermal lymphocytic infiltrate associated with melanophages. (**E**) Melan-A/MART-1 (original magnification 10x) of the same lesion, showing the same features and some pagetoid spread. (**F**) PRAME (original magnification 10x) is negative. (**G**) Hematoxylin and eosin stain (original magnification 10x). Superficial borderline atypical melanocytic lesion with junctional atypia. The presence of a very disorganized intraepidermal melanocytic proliferation with numerous solitary units and voluminous and irregular nests, with (inset) marked/severe cytonuclear atypia (abundant cytoplasm, and hyperchromatic nuclei; original magnification 30x). There is papillary fibroplasia, and a slight dermal lymphocytic infiltrate associated with melanophages. (**H**) Melan-A/MART-1 IHC (original magnification 10x) of the same lesion, showing the same features and pagetoid spread. (**I**) PRAME IHC (original magnification 10x) is negative. We discussed, as a differential diagnosis, the possibility of a melanoma in situ arising on a pre-existing dysplastic nevus in this case.

**Figure 3 dermatopathology-12-00003-f003:**
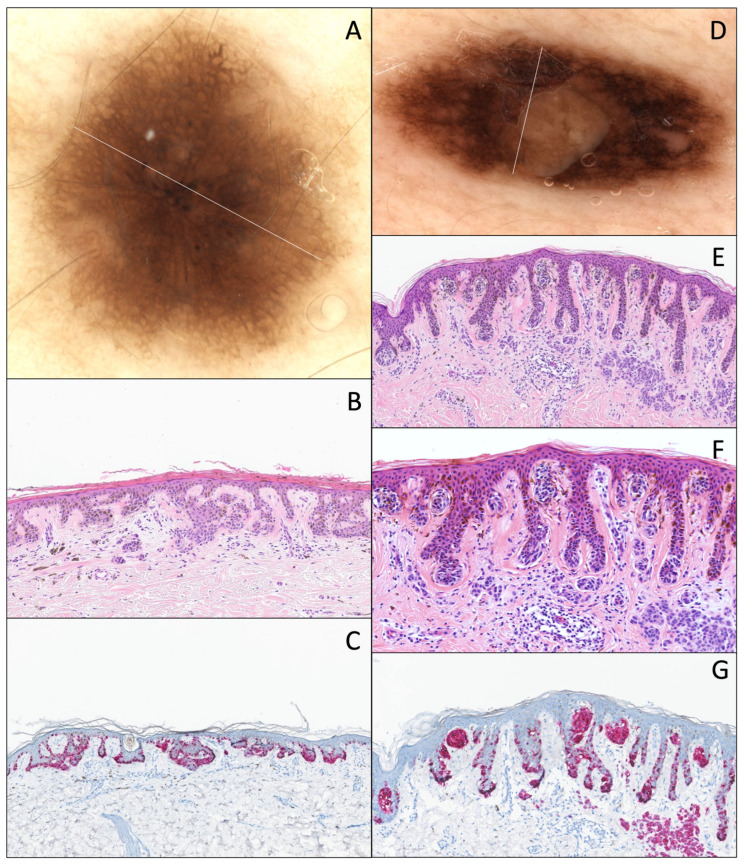
DN with mild atypia. (**A**) Dermatoscopy (original magnification 10x). The lesion is displaying relative symmetry and a thin pigmented reticular network with very few dots at the center. (**B**) Hematoxylin and eosin stain (original magnification 10x). Pigmented parakeratosis overlying a hyperplastic epidermis with lentigo-like features, associated with a melanocytic proliferation showing a discrete junctional proliferation with very small nests, irregularly distributed along the dermal–epidermal junction. There is a “shoulder sign”, a slight papillary fibroplasia, and a slight dermal lymphocytic infiltrate associated with melanophages. (**C**) Melan-A/MART-1 IHC (original magnification 10x) of the same lesion, showing the same features and the absence of continuous proliferation and pagetoid spread. DN with moderate atypia. (**D**) Dermatoscopy (original magnification 10x). The lesion shows a relative symmetry, an irregular pigmented reticular network, with foci showing a larger and more pigmented network, and a central area corresponding to a dermal component. (**E**) Hematoxylin and eosin stain (original magnification 10x). Junctional proliferation shows small nests irregularly distributed along the dermal–epidermal junction, not completely filling the rete ridges, and (**F**) with mild cytonuclear atypia (original magnification 15x). There is papillary fibroplasia, a “shoulder sign”, and a slight dermal lymphocytic infiltrate associated with melanophages. The dermal component shows a superficial congenital architecture. (**G**) Melan-A/MART-1 IHC (original magnification 10x) of the same lesion, showing the same features and the absence of continuous proliferation and pagetoid spread.

**Figure 4 dermatopathology-12-00003-f004:**
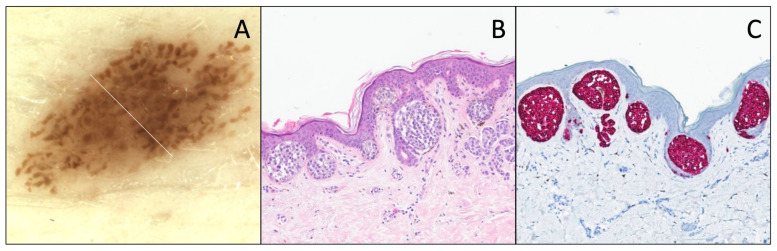
DN with distinct atypia, nested type. (**A**) Dermatoscopy (original magnification 10x). The lesion is displaying relative symmetry, an irregular pigmented reticular network, with many large dots. (**B**) Hematoxylin and eosin stain (original magnification 10x). Junctional proliferation shows large and voluminous nests irregularly distributed along the dermal–epidermal junction, completely filling the rete ridges, with moderate cytonuclear atypia. A slight papillary fibroplasia, a “shoulder sign”, and a slight dermal lymphocytic infiltrate associated with melanophages are observed. (**C**) Melan-A/MART-1 IHC (original magnification 10x) of the same lesion, shows the same features with no continuous proliferation or pagetoid spread.

**Figure 5 dermatopathology-12-00003-f005:**
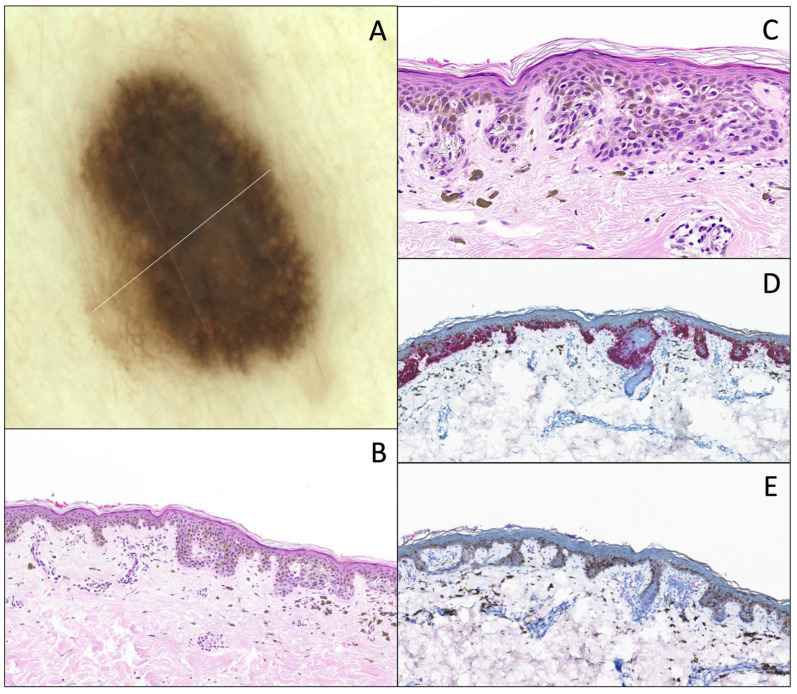
DN with marked atypia. (**A**) Dermatoscopy (original magnification 10x). The lesion shows a slight asymmetry, an irregular and coarse pigmented reticular network, with an abrupt/sharp interruption and a slight blue-gray color in its center part. (**B**) Hematoxylin and eosin stain (original magnification 10x). Junctional component shows a continuous proliferation of solitary units along the dermal–epidermal junction, and some irregular nests, with (**C**) marked cytonuclear atypia (large cytoplasm, and large nuclei with hyperchromasia; original magnification 20x). There is a slight papillary fibroplasia and a moderate dermal lymphocytic infiltrate associated with melanophages. (**D**) Melan-A/MART-1 (original magnification 10x) of the same lesion, showing the same features and very little pagetoid spread. (**E**) PRAME IHC (original magnification 10x) is negative.

**Figure 6 dermatopathology-12-00003-f006:**
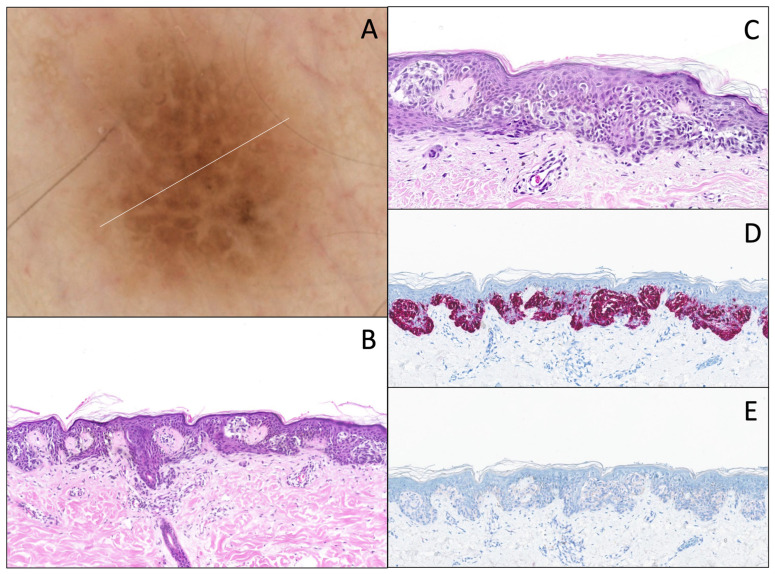
DN with severe atypia. (**A**) Dermatoscopy (original magnification 10x). The lesion shows a slight asymmetry, an irregular and somewhat “inverted” network made of diffuse white lines. (**B**) Hematoxylin and eosin stain (original magnification 10x). Junctional component shows a very disorganized proliferation with solitary units and irregular nests, with (**C**) marked/severe cytonuclear atypia (abundant cytoplasm, and large and irregular nuclei with hyperchromasia; original magnification 20x). There is a “shoulder sign”, papillary fibroplasia, and a moderate dermal lymphocytic infiltrate associated with melanophages. (**D**) Melan-A/MART-1 IHC (original magnification 10x) of the same lesion, showing the same features and very little pagetoid spread. (**E**) PRAME IHC (original magnification 10x) is negative.

**Figure 7 dermatopathology-12-00003-f007:**
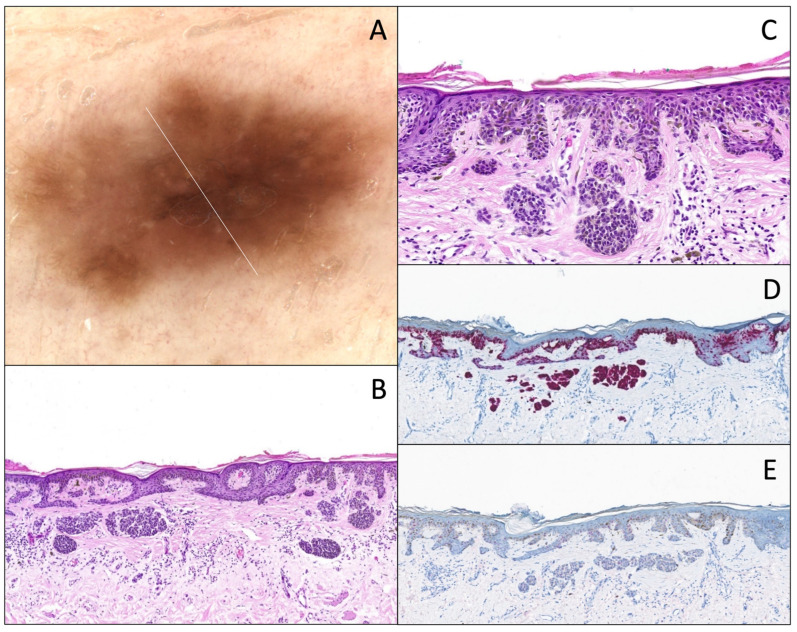
Superficial borderline atypical melanocytic lesion with junctional atypia. (**A**) Dermatoscopy (original magnification 10x). The lesion has features seen in melanoma such as asymmetry, multi-component pattern and multiple colors with a foci of coarse pigmented reticular network, and area without structure. (**B**) Hematoxylin and eosin stain (original magnification 10x). Junctional component shows a continuous proliferation with solitary units along the dermal–epidermal junction, with (**C**) severe cytonuclear atypia (vacuolated cytoplasm, and irregular nuclei with pronounced hyperchromasia; original magnification 20x). There is a “shoulder sign”, a papillary fibroplasia, and a slight dermal lymphocytic infiltrate associated with melanophages. (**D**) Melan-A/MART-1 IHC (original magnification 10x) of the same lesion, showing the same features and a little pagetoid spread. (**E**) PRAME IHC (original magnification 10x) is very focally positive (<25% of the junctional component).

**Figure 8 dermatopathology-12-00003-f008:**
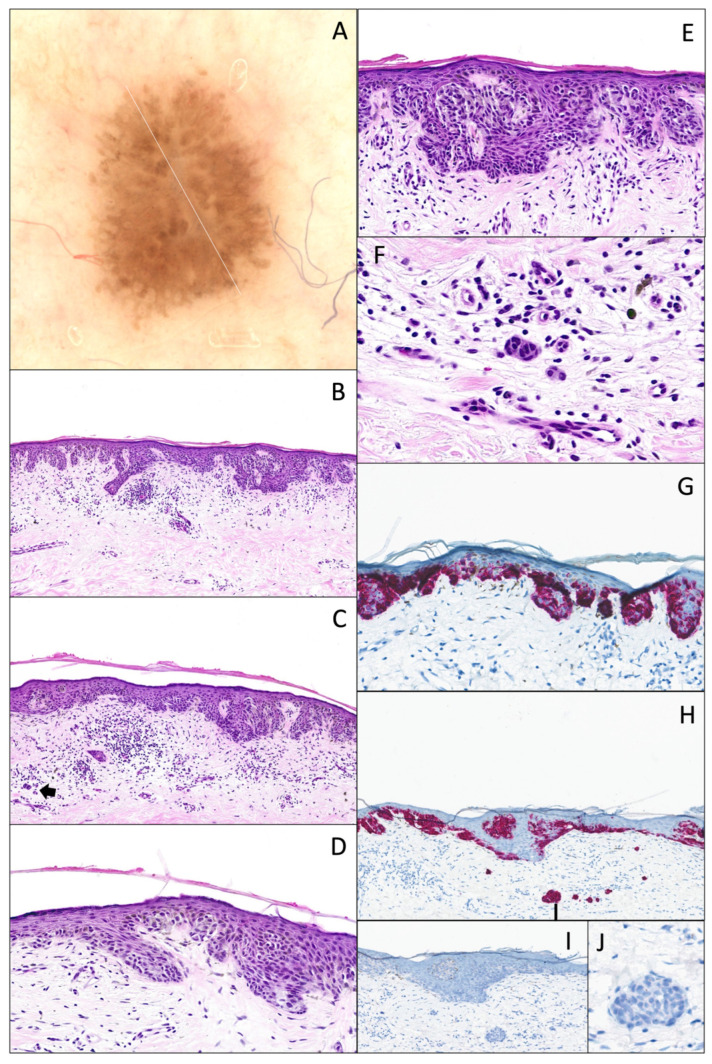
Superficial borderline atypical melanocytic lesion with junctional and dermal atypia. (**A**) Dermatoscopy (original magnification 10x). Presence of pseudopods, radial striae, an irregular pigmented network, and irregular large dots. (**B**) Hematoxylin and eosin stain (original magnification 10x). Junctional component shows a continuous proliferation with solitary units along the dermal–epidermal junction, with severe cytonuclear atypia (abundant and dusty cytoplasm, and irregular nuclei with pronounced hyperchromasia (original magnification 20x) (**D**,**E**). There is a “shoulder sign”, papillary fibroplasia, and a slight dermal lymphocytic infiltrate associated with melanophages (**C**). (**C**,**F**) The dermal component is focal (arrow), composed of few nests, showing the same degree of atypia, without differentiation, and with overlying fibrosis (original magnification 10x and 40x). (**G**,**H**) The same lesion with MelanA/MART-1 IHC (original magnification 10x), showing a continuous proliferation and pagetoid spread, and (**I**,**J**) PRAME IHC (original magnification 10x and 40x) is completely negative.

**Figure 9 dermatopathology-12-00003-f009:**
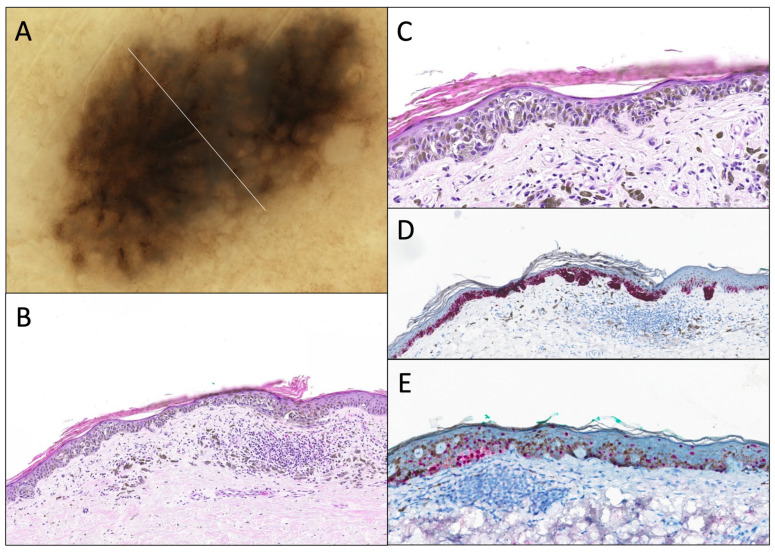
Melanoma in situ. (**A**) Dermatoscopy (original magnification 10x). The lesion shows an asymmetry, and an irregular and very coarse pigmented reticular network, with gray areas and radial striae. (**B**) Hematoxylin and eosin stain (original magnification 10x). Intraepidermal anarchic proliferation with irregular nests, many solitary units, and pagetoid spread, showing (**C**) severe cytonuclear atypia (very large cytoplasm, and very large nuclei; original magnification 20x. There is a moderate dermal lymphocytic infiltrate associated with melanophages. (**D**) Melan-A/MART-1 IHC (original magnification 10x) of the same lesion, showing the same features. (**E**) PRAME IHC (original magnification 15x) is diffusely positive.

**Figure 10 dermatopathology-12-00003-f010:**
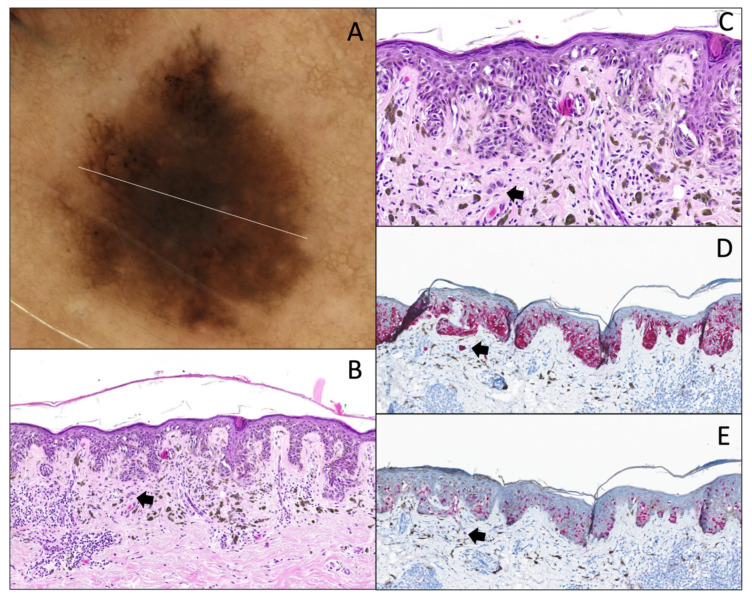
Superficial spreading melanoma (Breslow 0.3 mm). (**A**) Dermatoscopy (original magnification 10x). The lesion shows an asymmetry, multi-component pattern, areas without structure, irregularly distributed dots and an abrupt interruption. (**B**) Hematoxylin and eosin stain (original magnification 10x). Intraepidermal anarchic proliferation with irregular nests, many solitary units and pagetoid spread, showing (**C**) severe cytonuclear atypia (very large cytoplasm, and very large nuclei), with the same atypia in few nests in the superficial dermis (arrow). There is a moderate dermal lymphocytic infiltrate associated with melanophages (original magnification 20x). (**D**) Melan-A/MART-1 IHC (original magnification 10x) of the same lesion, highlighting the solitary units and the pagetoid spread, as well as the presence of very few nests in the superficial dermis (arrow). (**E**) PRAME IHC (original magnification 10x) is positive (>75% with strong positivity), including the dermal component (arrow).

**Figure 11 dermatopathology-12-00003-f011:**
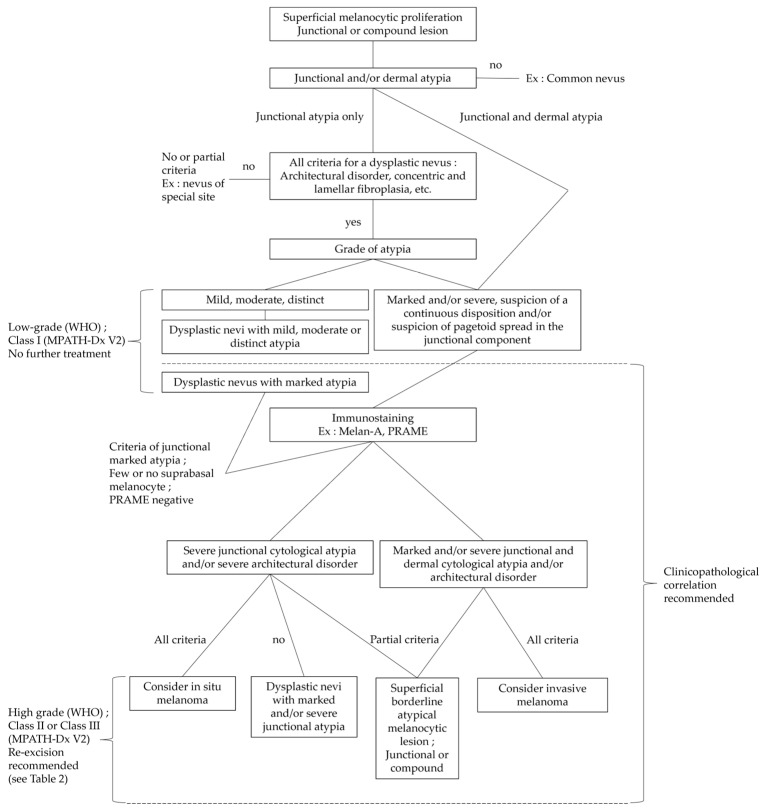
Algorithmic classification of superficial melanocytic proliferations.

**Table 1 dermatopathology-12-00003-t001:** Our classification and the associated clinical/dermatoscopic features.

Classification	Histologic Features/Criteria	Clinical/Dermatoscopic Features
DN with mild atypia	Cytonuclear criteria: - Mild cytonuclear atypia: - Nuclear size ≤ keratinocyte nucleus; - Slight hyperchromatism; - Nucleolus absent or small; - Little cytoplasm. Architectural criteria: - Lentigo-like epidermal hyperplasia and hyperpigmentation often present; - Discrete junctional proliferation showing very small nests, irregularly distributed at the dermal–epidermal junction.	Thin reticular pigmented network in the majority of the lesion
DN with moderate atypia	Cytonuclear criteria: - Mild cytonuclear atypia: - Nuclear size ≤ keratinocyte nucleus; - Slight hyperchromatism; - Nucleolus absent or small; - Little cytoplasm. Architectural criteria: - Larger nests, not completely filling the rete ridges.	Slightly larger pigmented network and/or small globi
DN with distinct atypia	Cytonuclear criteria: - Moderate cytonuclear atypia: - Nuclear size ≈ 1.5x keratinocyte nucleus; - Slight pleomorphism; - Nucleolus absent or small; - Abundant cytoplasm, sometimes with dusty pigmentation. Architectural criteria: - Voluminous nests, or solitary units, completely filling the rete ridges; - No suprabasal melanocyte proliferation and no solitary units between the rete ridges.	Larger and coarser pigmented network and/or larger globi, sometimes at the periphery of the lesion
DN with marked atypia	Cytonuclear criteria: - Marked or severe cytonuclear atypia: - Nuclear size ≥ 2x keratinocyte nucleus; - Severe hyperchromatism, or vesicular nucleus; - Nucleolus prominent or enlarged; - Abundant and dusty cytoplasm. Architectural criteria: - Continuous proliferation along the dermal–epidermal junction, or with solitary units between the rete ridges; - Very few suprabasal melanocytes, only focally.	May exhibit severe dermatoscopic features such as: radial striae, pseudopods, inverted network, eccentric area without structure, asymmetry, multi-component pattern These features are also seen in DN with severe atypia, superficial borderline atypical melanocytic lesions and melanoma
DN with severe atypia	Cytonuclear criteria: - Marked or severe cytonuclear atypia: - Nuclear size ≥ 2x keratinocyte nucleus; - Severe hyperchromatism, or vesicular nucleus; - Nucleolus prominent or enlarged; - Abundant and dusty cytoplasm. Architectural criteria: - Continuous proliferation along the dermal–epidermal junction, or with solitary units between the rete ridges; - Many suprabasal melanocytes, generally restricted to the central part of the lesion; - The lesion does not fulfill the criteria of a melanoma in situ.	May exhibit severe dermatoscopic features such as radial striae, pseudopods, inverted network, eccentric area without structure, asymmetry, and multi-component pattern These features are also seen in DN with marked atypia, superficial borderline atypical melanocytic lesions, and melanoma
Superficial borderline atypical melanocytic lesion—junctional atypia	Cytonuclear criteria: - Cytoplasmic atypia: dirty cytoplasm, epithelioid morphology, vacuolated cytoplasm; - Nuclear atypia: large nucleus, hyperchromasia; - Nucleolar atypia: prominent nucleolus. Architectural criteria: - Disorganized/anarchic proliferation with irregular nests and numerous solitary units; - Some pagetoid spread. These atypia/criteria are not observed throughout the lesion; otherwise, the lesion fulfills the criteria of a melanoma in situ.	May exhibit severe dermatoscopic features such as radial striae, pseudopods, inverted network, eccentric area without structure, asymmetry, and multi-component pattern These features are also seen in DN with marked and severe atypia, and melanoma
Superficial borderline atypical melanocytic lesion—junctional and dermal atypia	The same criteria as above apply for the intraepidermal component, associated with dermal atypia: - Same degree of cytonuclear atypia (dirty cytoplasm, epithelioid morphology, large nucleus, hyperchromasia, etc.); - Absence of maturation; - Eventually few mitotic figures.	May show severe dermatoscopic features such as radial striae, pseudopods, inverted network, eccentric area without structure, asymmetry, and multi-component pattern These features are also seen in DN with marked and severe atypia, and melanoma
Melanoma in situ (SSM subtype)	Severe cytonuclear and architectural atypia throughout the lesion. Cytonuclear criteria: - Usually epithelioid morphology; - Nuclear atypia: very large nucleus, angulous shape, severe hyperchromasia; - Nucleolar atypia: very prominent nucleolus. Architectural atypia: - Disorganized/anarchic proliferation with irregular nests and numerous solitary units; - Intense pagetoid spread (“buckshot” pattern); - Large lesion.	Usually exhibits severe dermatoscopic features such as radial striae, pseudopods, an inverted network, eccentric area without structure, blue-white areas, etc.
Superficial spreading melanoma	Severe cytonuclear and architectural atypia throughout the lesion. Cytonuclear criteria: - Epithelioid morphology; - Nuclear atypia: very large nucleus, angulous shape, severe hyperchromasia; - Nucleolar atypia: very prominent nucleolus. Architectural atypia: - Disorganized/anarchic proliferation with irregular nests and numerous solitary units; - Intense pagetoid spread (“buckshot” pattern); - Large lesion; - Asymmetry. Dermal component: - Usually the same degree of cytonuclear atypia is seen in the totality of the dermal component; - Total absence of maturation; - Presence of mitotic figures; - Asymmetry; - Expansile nests; - Sometimes many different morphologies are seen: epithelioid, spindle cells, clear cells, pigmented cells, nevoid cells, etc.; - Regression, ulceration.	Usually exhibits severe dermatoscopic features such as radial striae, pseudopods, an inverted network, eccentric area without structure, blue-white areas, etc.

**Table 2 dermatopathology-12-00003-t002:** Comparative table. Our classification and MPATH dx V2, with the associated management [29].

Our Classification	Management	MPATH Dx V2	Management
DN with mild atypia	No further treatment	Class I: low-grade	No further treatment
DN with moderate atypia	No further treatment	Class I: low-grade	No further treatment
DN with distinct atypia	No further treatment	Class I: low-grade	No further treatment
DN with marked atypia	No further treatment	Class I: low-grade	No further treatment
DN with severe atypia	Re-excision with 5 mm margin	Class II: high-grade	Re-excision with margins < 1 cm
Superficial borderline atypical melanocytic lesion with only junctional atypia	Re-excision with 5 mm margin	Class II: high-grade	Re-excision with margins < 1 cm
Superficial borderline atypical melanocytic lesion with junctional and dermal atypia	Re-excision with 5–10 mm margin	Class II: high-grade	Re-excision with margins < 1 cm
Melanoma in situ	Re-excision with 5 mm margin	Class II: high-grade	Re-excision with margins < 1 cm
Superficial spreading melanoma (Breslow < 0.8 mm)	Follow national guidelines (e.g., wide excision with 1 cm margins)	Class III: melanoma pT1a	Follow national guidelines (e.g., wide excision with 1 cm margins)

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
