# Peer review of "Dysplastic Nevi and Superficial Borderline Atypical Melanocytic Lesions: Description of an Algorithmic Clinico-Pathological Classification"

_dermatopathology, 2025, doi:10.3390/dermatopathology12010003_

Round 1
Reviewer 1 Report
Comments and Suggestions for Authors
Congratulations to the authors for this interesting and very useful paper titled "Dysplastic nevi and borderline atypical melanocytic lesions: Description of an algorithmic clinico-pathological classification": I have learned so much! Firstly, the authors reported about the important question of the "gray zone" for the correct nosographic classification of the Dysplastic Nevus (DN) and the early Malignant Melanoma (MM), on the light of the current classification system such as MPATH-Dx V.2. I believe that the introduction of more categories can always add other confusing factors but the authors clearly stated the their goal is to present to scientific community this approach. Secondly, the pictures are very beautiful to see, with clinical iconography that allow to the readers to appreciate the correlation between the two different points of view (clinical and histological). Thirdly, in the discussion section, authors performed a deep analysis of their scheme in order to other different classification systems.
To finish, I think that this paper HAVE TO published because it can help many dermatopathologists, pathologists and physicians.
Only two minor comments: line 235: not "ND" but always "DN" and in the caption of Figure 10: not "E" but "D" and not "F" but "E" figures.
Reviewer 2 Report
Comments and Suggestions for Authors
The article addresses a very current and always controversial topic: the definition of dysplastic nevus.
The authors have submitted a work that reviews the literature on dysplastic nevus and also provides a classification (see paragraph 2 and following).
The article and classification explain in detail, including the architectural and cytological criteria, by which a melanocytic lesion with atypia can be classified as: DN with mild atypia, moderate atypia, distinct atypia, marked atypia, and severe atypia.
I believe that when a lesion is evaluated by a dermopathologist, it is necessary and useful to classify it into different clinical entities (each representing a different tone on a grayscale).
This distinction is useful from a didactic point of view (for both pathologists and, especially, dermatologists), but less so in clinical practice. In fact, the World Health Organization released the new Classification of Skin Tumours in 2018, with updated recommendations on grading dysplasia in dysplastic nevi.
Dysplastic nevi are now graded as “low grade dysplastic nevus” or “high grade dysplastic nevus.”
- How should DN with moderate, distinct, and marked atypia be classified? Should they be classified as low grade or high grade? According to Table 3 (comparative), it seems that DN with mild, moderate, distinct, and marked atypia are equated to low grade in the MPATH dx V2 classification. However, this is unclear in the text.
A comment: If we adopt a classification that divides dysplastic nevi into mild/moderate/distinct/marked and severe, we risk promoting a system that increases intra- and inter-operator diagnostic differences, while reducing reproducibility. Ultimately, I think diagnoses should converge into DN with low grade or high grade, even in the classification you presented. - I would avoid using the term “borderline atypical melanocytic lesions” in the title, text, or tables.
The article discusses “intermediate” lesions but almost exclusively refers to the “classical” subgroup (superficial/thin compound: dysplastic nevus). - I would avoid mentioning AST, PEM, STUMP, and DPN in the introduction and discussion without sufficient explanation. These belong to the “non-classical” subgroup (thick compound/dermal: “melanocytomas”), which has genetic pathways distinct from classical melanoma subtypes. In fact, the large family of melanocytomas has clinical, dermatoscopic, and histological characteristics (mainly dermal, deep lesions) that differ from those of DN and “classical” melanoma.
I would like to specify that when I refer to the “classical” subgroup (superficial/thin compound: dysplastic nevus) and “non-classical” subgroup (thick compound/dermal: “melanocytomas”), I am referencing the subdivision described in the following article: Ferrara G, Argenziano G. The WHO 2018 Classification of Cutaneous Melanocytic Neoplasms: Suggestions From Routine Practice. Front Oncol. 2021 Jul 2;11:675296. doi: 10.3389/fonc.2021.675296. PMID: 34277420; PMCID: PMC8283700.
In this article, I would focus only on the “classical” subgroup (superficial/thin compound: dysplastic nevus), as the “non-classical” subgroup (thick compound/dermal: “melanocytomas”) represents another grayscale that warrants a separate review article.
Table 2 has the following issues:
- The final diagnosis of dysplastic nevus should ultimately be classified as either low or high grade (as indicated by the WHO 2018 classification).
- “Clinicopathological correlation recommended” should not be placed at the end of the flowchart in Table 2. Logically, it should come before the diagnoses of melanoma in situ, DN with marked/severe atypia, borderline atypical melanocytic lesions, and invasive melanoma.
- In the same box as DN with severe atypia, DN with marked junctional atypia is also included. However, the latter is classified as low grade in Table 3, which is somewhat confusing.
In general, the strengths of the article are definitely the dermatoscopic and pathological clinical correlation images, which serve as a sort of illustrated “atlas” on DN, melanoma in situ, and SSM. I agree on the importance of providing a diagnosis after correlating the clinical and pathological aspects of each lesion.
Comments on the Quality of English Language
The English language needs substantial improvement. The sentences are generally long and complex, and there are typographical errors (e.g., in the table, “asymetry” instead of “asymmetry” referring to SSM). In some places, the reading is not very fluid.
Reviewer 3 Report
Comments and Suggestions for Authors
Thank you for allowing me to read and review this interesting paper.
As the authors highlight in the introduction, the concept of dysplastic nevus remains one of the most challenging and controversial areas in melanocytic pathology and requirs more substantive data in the literature to be effectively employed by all histopathologists.
One of the most important limit in the diagnosis and classification of dysplastic nevi is, that often rely on subjective criteria, which can lead to variability and reproducibility issues. The subjective nature of the classification complicates efforts to establish consensus or standardization.
The authors' effort to propose a new classification for dysplastic nevi is praiseworthy, as it aims to provide clarity on this complex issue. The proposed classification is well-illustrated, with detailed definitions and excellent figures that help with the reader's understanding of the proposed criteria.
However, several concerns arise from this new approach:
-
The proposed classification's division of dysplastic nevi into five distinct categories, in my opinion, may increase complexity. While the framework is clearly articulated, it still risks reducing the reproducibility of the diagnostic process. Particularly, distinguishing between "distinct" and "marked" atypia in everyday clinical practice can be challenging and could lead to significant variability among observers.
-
The paper, in my opinion, does not adequately address how this new classification benefits clinical management or patient outcomes. Without a clear discussion of the practical implications or advantages for clinicians and patients, the utility of this classification remains ambiguous.
-
Incorporating molecular analyses, if available, could increase the reproducibility, and credibility of the proposed classification. Molecular data could provide an additional layer of objectivity that morphology alone cannot achieve.
-
The paper would benefit from a clearer statement of its overall objectives. The introduction and conclusion sections should be expanded to provide a more explicit explanation about the study's mission and how the proposed classification relates to broader efforts to enhance the diagnosis and management of melanocytic lesions.
In conclusion, while the paper makes a valuable contribution to the ongoing dialogue around dysplastic nevi by proposing a well-defined classification scheme with illustrative examples, it raises some concerns about the practicality and applicability of this approach.
Clarifying the clinical implications, addressing the reproducibility concerns, and integrating molecular data could greatly enhance the value and impact of this study.
Therefore, I would strongly encourage the authors to refine their introduction and conclusion to provide a more comprehensive understanding of the paper's purpose and potential contributions to the field.
Round 2
Reviewer 2 Report
Comments and Suggestions for Authors
The authors have responded to all comments clearly and comprehensively. I have no comments to add.
Reviewer 3 Report
Comments and Suggestions for Authors
As the authors mentioned, if the purpose of this is "the study aims to have an educational aspect for the dermatopathologists and dermatologists using dermatoscopy in routine practice" I would accept this last version of the paper since I believe the paper is well written and interesting. However, I am still very concerned that this classification can be used in the practical routine diagnosis of the melanocytic lesions.